# Evaluation of Antimicrobial Activity in the Extract of Defatted *Hermetia illucens* Fed Organic Waste Feed Containing Fermented Effective Microorganisms

**DOI:** 10.3390/ani12060680

**Published:** 2022-03-08

**Authors:** Kyu-Shik Lee, Eun-Young Yun, Tae-Won Goo

**Affiliations:** 1Department of Pharmacology, College of Medicine, Dongguk University, Gyeongju 38766, Korea; there1@dongguk.ac.kr; 2Department of Integrative Bio-Industrial Engineering, Sejong University, Seoul 05006, Korea; yuney@sejong.ac.kr; 3Department of Biochemistry, College of Medicine, Dongguk University, Gyeongju 38766, Korea

**Keywords:** antimicrobial peptide, *Hermetia illucens* larvae, natural antimicrobial substance, ecofriendly feedstock, food additives

## Abstract

**Simple Summary:**

We assessed the antimicrobial activity of *Hermetia illucens* larvae (HIL) extract from crude-oil-extracted crushed powder prepared by using a developed automatic oil extractor for biodiesel production. We found the extract effectively reduced the survival of pathogens and antimicrobial-peptide-resistant bacteria. The results demonstrate that defatted HIL extract from crude-oil-extracted crushed powder prepared by using a developed automatic oil extractor should be used as a feed additive having antimicrobial activity with low production cost.

**Abstract:**

*Hermetia illucens* (black soldier fly) larvae (HIL) are considered useful industrial insects for the production of feed for livestock, eco-friendly fertilizer from organic wastes, and biodiesel. Therefore, we evaluated the antimicrobial activity in the extract of crude-oil-extracted crushed HIL powder prepared from HIL fed organic waste containing fermented effective micro-organisms for biodiesel production. The result showed that antimicrobial activity was not fully induced in HIL fed *L. casei*-containing feed. In contrast, increased antimicrobial activity was observed in defatted HIL extract prepared from crude-oil-extracted crushed HIL powder. We found that the extract effectively inhibited the growth of pathogens and antimicrobial-peptide-resistant bacteria, such as three kinds of *Salmonella* species, and *Enterococcus faecalis*, *Streptococcus mutans*, *Candida albicans*, *Serratia marcescens*, and *Pseudomonas tolaasii*, with a minimum inhibitory concentration of 200–1000 µg/100 µL. Furthermore, no cytotoxicity to CaCO-2 human intestinal cells was observed in the extract. We also found that the production fee of extract equivalent to the antimicrobial activity of melittin was approximately 25-fold less than the production fee of melittin. Therefore, the results demonstrate that crude-oil-extracted crushed HIL powder prepared from HIL fed organic waste containing fermented effective micro-organisms for biodiesel production should be used as the feedstock for synthetic, preservative-free livestock feed and food additives. Taken together, the present study supports the usefulness of HIL as an eco-friendly feedstock in the biodiesel, agricultural, food, and feed industries.

## 1. Introduction

Many investigators have reported on the use of black soldier fly larvae, *Hermetia illucens*, in the production of fertilizer, livestock feed, and fishery feed [1,2,3,4]. Furthermore, several investigations showed that *H. illucens* larvae (HIL) can be used as protein-rich food for humans [5]. HIL were also studied as a living factory for producing feedstock for natural fertilizer because HIL can bioconvert various organic wastes, including food waste, human waste, and animal manures, to valuable natural substances [6,7,8]. The investigations demonstrate that *Hermetia illucens* is a useful insect for agricultural and eco-friendly industries.

In modern times, the industrial farming of livestock has been established to improve livestock productivity because meat consumption has increased. Therefore, to improve productivity, intensive farming of livestock and animal growth promotion was required. The requirements were achieved by using antibiotics in animal feed. However, although antibiotics effectively prevent infectious diseases from developing, many problems, such as antibiotic resistance, antibiotic residues in livestock products, ecosystem pollution from livestock manure, and weak disease resistance in livestock, were caused by the abuse of in-feed antibiotics [9,10,11]. Therefore, the development of natural antimicrobial substances is necessary to solve the problems caused by the abuse of in-feed antibiotics.

Many researchers have investigated and searched for natural antibiotic substances and suggested that insects are potent natural antibiotic substances because insects produce antimicrobial peptides (AMPs) via activation of their innate immune system to prevent pathogen invasion [12,13,14,15]. Furthermore, insects have been considered an alternative protein source and suggested as an animal feed [16,17]. Many investigations suggested that HIL, grasshoppers, yellow mealworms, silkworms, and termites are strong candidates for industrial feed production [18]. Therefore, the suggestions imply that, due to their natural antibiotic activity, insects should be used as alternative animal feed.

Organic wastes are colonized by micro-organisms. Therefore, with HIL intake organic waste, microbes can be ingested into HIL. As a result, the innate immune response in HIL should be activated and can produce AMPs [19,20]. In previous works, we found that the gene expression of AMPs in HIL was enhanced by artificial infection of *Lactobacillus casei* using a fine needle [21]. Moreover, Park et al. purified and characterized AMPs from HIL [22], and some investigations demonstrated that AMPs were expressed and secreted when HIL were immunized by bacteria [23,24]. We also showed antimicrobial activity against *Salmonella* species in *L. casei*-immunized HIL extract with increased expression of AMPs genes, such as cecropin 1 and defensin 1 [25]. The investigations showed that AMPs gene expression and antimicrobial activity in insects were significantly enhanced by bacterial infection.

HIL have also been studied as a source of biodiesel because they contain a high content of lipids [26]. Several investigations evaluated the possibility of HIL as biodiesel feedstock and analyzed lipid composition [27,28,29]. Furthermore, Harlystiarini et al. showed that HIL extract contains a high level of lauric acid (49.18%), a saturated fatty acid having weak antimicrobial activity [30]. They also suggested that lauric acid contributed to the antimicrobial activity of HIL extract through the induction of AMP. In addition, antimicrobial activity is affected by various lipids and fatty acids [31]. The reports imply that HIL can be used for biodiesel production, and the antimicrobial activity in HIL extract should be lessened after lipid extraction because the lipid and fatty acid contents in HIL extract are decreased. Therefore, we assessed antimicrobial activity in defatted HIL extract to investigate whether the antimicrobial activity was affected by the lipid content of HIL extract and to evaluate whether the extract should be used as an antibiotic-free animal feed.

## 2. Materials and Methods

### 2.1. Hermetia illucens and Bacterial Strains

HIL were gifted from the Department of Agricultural Biology at the National Institute of Agricultural Sciences of the Rural Development Administration (Wanju, Korea). HIL were routinely incubated at room temperature (26 ± 1 °C) in relative humidity of 60%. We purchased various bacteria strains to assess the antimicrobial activity of HIL extract. *Escherichia coli* (KCCM 11234) and three *Lactobacillus* strains, namely *Lactobacillus brevis* (KCCM 10553), *Lactobacillus casei* (KCCM 12413), and *Lactobacillus fermentum* (KCCM 11441), were obtained from the Korean Collection for Type Culture (Wonju, Korea). The yeast *Candida albicans* (KACC 30071), the pathogenic bacteria *Enterococcus faecalis* (KACC 11859), *Streptococcus mutans* (KACC16833), *Salmonella pullorum* (KVCC-BA0702509), *Salmonella typhimurium* (KCCM 40406), and *Salmonella enteritidis* (KCCM 12021), and the AMP-resistant bacteria *Serratia marcescens* (KACC11961) and *Pseudomonas tolaasii* (KACC15293) were bought from the National Institute of Animal Science of the Rural Development Administration to assess the antibiotic activity of HIL extracts (Wonju, Korea). All experiments were performed in Biosafety Level-2 laboratories approved by the Korean Disease Control and Prevention Agency.

### 2.2. HIL Breeding

Six-day-old HIL were inoculated into feed prepared using a mixture of dried food waste (60 g/100 g), chicken manure (40 g/100 g), and waste cooking oil (2 mL/100 g) containing a fermented effective microbe (F-EM) solution (0.8%). F-EM is a fermented solution made by mixing cultures containing lactic-acid bacteria, photosynthesis bacteria, and yeast. HIL were grown for 11 days and harvested. The live HIL were put in a 700-W microwave and dried for 8.5 min.

### 2.3. Preparation of Defatted HIL Extract

Dried HIL were defatted by expeller press, and crushed HIL powder was collected and stored at −20 °C until use. Defatted, crushed HIL powder (50 g) was suspended in 500 mL of 0–30% acetic acid solution and boiled at 100 °C for 10, 30, or 60 min. Then, the crushed HIL powder suspension was centrifuged at 4500 rpm for 30 min at 4 °C, and then the supernatant was removed and 0.45 µm filtered. The filtered supernatant was collected and dried for 16 h in a vacuum-spin drier. The dried supernatant was resuspended in sterilized 0.05% acetic acid solution and centrifuged at 14,000 rpm for 10 min at 4 °C. The supernatant was transferred to a new tube and held as defatted HIL extract. The extract was stored at −20 °C.

### 2.4. Assessment of Antimicrobial Activity in Defatted HIL Extract

A radial-diffusion assay (RDA) was used to evaluate the antimicrobial activity of the defatted HIL extract against *E. coli* and *S. aureus*. Underlay gel (9 mM sodium phosphate, 1 mM sodium citrate, pH 7.4, 1% low-electroendosmosis agar, and 0.03% tryptic soy broth) was sterilized and liquefied by autoclave, mixed with each type of bacteria after cooling to 50–55 °C and poured onto a 100-mm square plate. After the underlay gel hardened, 3.5-mm diameter wells were prepared on the underlay gel, and then the defatted HIL extract was placed into the wells and held at 37 °C. After 3-h incubation, a sterilized overlay gel (6% tryptic soy broth and 1% low-electroendosmosis agar) was used to cover the underlay gel, and the gel plates were incubated for 18 h at 37 °C. Then, we determined antimicrobial activity of defatted HIL extract by measuring clear zone widths. We used melittin (Sigma-Aldrich; Merck KGaA, Darmstadt, Germany) as a positive control.

### 2.5. Determination of Minimum Inhibitory Concentrations (MICs) of Defatted HIL Extract

Antimicrobial activity of defatted HIL extract against bacteria was measured as MICs. To assess MICs, bacteria were grown in liquid media to 4 × 10^6^ cfu/mL at 37 °C with 200 rpm shaking. Then, bacterial cultures were diluted to 1 × 10^6^ cfu/mL, and 90 µL of the diluent was added to each well of a 96-well plate. The diluted bacterial cultures were incubated for 18 h at 37 °C after treatment with serially diluted extracts (10 µL). MICs were determined by normalized absorbance of bacterial cultures at 600 nm. We measured the absorbance of diluted extract solutions for each concentration to normalize absorbance in bacterial cultures treated at the same concentration.

### 2.6. Analysis of Cytotoxicity of Defatted HIL Extract to Caco-2 Human Intestinal Cell

We analyzed the cytotoxicity of defatted, crushed HIL powder extract using Caco-2 human intestinal cells using 3-(4,5-dimethylthiazol-2-yl)-2,5-diphenyltetrazolium bromide (MTT) reagent. We plated 1 × 10^4^ cells into 96-well plates and incubated cells to attach for 24 h in a 5% CO_2_ atmosphere at 37 °C. Then, culture medium was replaced with conditioned media containing various concentrations of defatted, crushed HIL powder extract (0–5000 µg/100 µL) and further incubated for 24 h. After the incubation, 20 µL of 5 mg/mL MTT reagent was administrated to each well and left for 4 h in the dark. Then, 200 µL of dimethyl sulfoxide was added to each well to dissolve the formazan after removing the conditioned media. The absorbance of formazan dissolved in dimethyl sulfoxide was measured at 540-nm wavelength using an ELISA reader.

### 2.7. Statistical Analysis

Statistical analysis of data was performed by the one-way analysis of variance followed by Fisher’s LSD test or Student *t*-test using SPSS Ver. 20.0 (SPSS Inc., Chicago, IL, USA). A significant difference for each experimental group against control was considered when the *p* value was less than 0.05. All values were represented as means ± standard deviations (SD).

## 3. Results

### 3.1. Evaluation of Antimicrobial Activity in HIL Extract Produced from Crude-Oil-Extracted Crushed HIL Powder

Firstly, we assessed the nutritional composition of the HIL and crushed HIL powder (Appendix A). The result showed that almost all lipids were removed by the expeller extractor. Then, we evaluated the antimicrobial activity of the defatted HIL extract produced from the crude-oil-extracted crushed HIL powder that was prepared from HIL fed organic waste containing F-EM. The result showed that antimicrobial activity was closely linked with the treated acetic acid concentration during the process of preparing the HIL extract from the crushed HIL powder (Figure 1). Moreover, the highest antimicrobial activity was shown when crushed HIL powder was treated with 20% acetic acid solution at 100 °C for 30 min, and no antimicrobial activity was observed in the extract prepared from autoclaved-water-treated crushed HIL powder. This result demonstrated that a 20% acetic acid solution is suitable for preparing defatted HIL extract from crude-oil-extracted crushed HIL powder.

### 3.2. Quantification of Antimicrobial Substance in Defatted HIL Extract

We also assessed the yield of defatted HIL extract from crude-oil-extracted crushed-HIL powder. The result showed 150 g of defatted HIL extract could be prepared from 1 kg of crude-oil-extracted crushed HIL powder. Then, we quantified the content of the antimicrobial substance in the HIL extract to evaluate the economic value. At first, the antimicrobial activity of the defatted HIL extract (5000 µg) was measured by RDA and compared to the activities of melittin using *E. coli* and *S. aureus*. Then, we obtained the standard curve of the inhibitory zone of melittin to determine the relative content of the antimicrobial substance in HIL extract compared with melittin (Figure 2A,B). The result showed that the antimicrobial activity of 5000 µg HIL extract was similar to that of 19.7 µg melittin against *E. coli* and to that of 21.3 µg of melittin against *S. aureus*. This means 1 kg of HIL extract contains antimicrobial activity equivalent to 2.3 g of melittin against *E. coli* and 2.45 g of melittin against *S. aureus*.

### 3.3. Effect of Defatted HIL Extract on Pathogenic Bacterial Survival

Based on the above results, the antimicrobial activity of defatted HIL extract on pathogenic bacteria survival was estimated by determining the MICs. Firstly, we evaluated the effect on the survival of *E. coli*, a Gram-negative bacterium, and *S. aureus*, a Gram-positive bacterium. The result showed that the MIC for *E. coli* and *S. aureus* was 300 µg/100 µL (Figure 3). Next, we assessed the antimicrobial activity of HIL extract on pathogenic micro-organisms, such as *Enteroccocus faecalis*, *Streptococcus mutans*, *Salmonella pullorum*, *Salmonella typhimurium*, *Salmonella enteritidis*, and *Candida albicans*. HIL extract effectively inhibited the survival of *S. marcescens*, *S. pullorum*, and *S. typhimurium* in a MIC range of 200–300 µg/100 µL (Figure 4A), and the MICs of HIL extract against *E. faecalis*, *S. mutans*, and *C. albicans* were 200 µg/100 µL, 1000 µg/100 µL, and 1000 µg/100 µL, respectively (Figure 4B). These results indicate that defatted HIL extract is a potent antibiotic substance for preventing the survival of pathogenic Gram-negative and Gram-positive bacteria and *C. albicans*.

### 3.4. Effect of Defatted HIL Extract on AMP-Resistant Bacterial Survival

AMP is a strong candidate to overcome antibiotic-resistant bacteria. However, some bacteria can recover from the cytotoxicity of AMPs. Therefore, we evaluated the antimicrobial activity of defatted HIL extract against *S. marcescens* and *P. tolaasii*, AMP-resistant bacteria, by determining the MIC. The result showed that 1–10 µg/100 µL of melittin did not affect the survival of *S. marcescens* and *P. tolaasii* (Figure 5). In contrast, defatted HIL extract reduced the survival of *P. tolaasii* with a MIC of 200 µg/100 µL, and the viability of *S. marcescens* was significantly decreased at a MIC of 300 µg/100 µL (Figure 5). This result demonstrates that defatted HIL extract is an effective substance for preventing the growth of AMP-resistant bacteria in animal and fishery feed.

### 3.5. Effect of Defatted HIL Extract on Lactobacillus Species and Normal Animal Cell Survival

Although defatted HIL extract has strong antimicrobial activity against pathogens, if the cytotoxicity to probiotics, such as *Lactobacillus* species, is higher, it cannot be used as a feed additive. Therefore, we assessed the cytotoxicity of defatted HIL extract to *Lactobacillus* species such as *L. brevis*, *L. casei*, and *L. fermentum*. As shown in Figure 6A, all *Lactobacillus* species were killed by 1 µg/100 µL of melittin. In contrast, the MICs of defatted HIL against *Lactobacillus* species were 1000 or 2000 µg/100 µL (Figure 6A). The MIC to Lactobacillus species was approximately five-fold higher than the MIC to the pathogen (Figure 5 and Figure 6A). We also estimated the cytotoxicity of defatted HIL extract to normal animal Caco-2 cells. The result showed that although 5000 µg/100 µL of defatted HIL extract inhibited the cell viability of Caco-2 cells (inhibition ratio is 57%), no cytotoxicity was observed at less than 4000 µg/100 µL of defatted HIL extract (Figure 6B). Interestingly, the cell viability was higher than that of the control (0 µg/100 µL). This means that the extract may contain nutrients to enhance cell growth because the nutrients were extracted from crude-oil-extracted HIL powder. Consequently, these results indicate that defatted HIL extract should be used as antimicrobial material to prevent pathogen infection in animals without cytotoxicity to probiotics and animal cells.

## 4. Discussion

Here, we showed the highest antimicrobial activity was observed in defatted HIL extract manufactured with a 20% aqueous acetic acid solution (Figure 1). In contrast, the lowest antimicrobial activity was shown when crushed HIL powder was extracted with distilled water. Although an investigation showed that antimicrobial activity was positively affected by the content of lauric acid, a saturated fatty acid with a 12-carbon atom chain, in HIL extract [30], Vogel et al. showed higher antimicrobial activity in the chloroform extract of black-soldier-fly-fed, micro-organism-contaminated feed than that of water extract [32]. Furthermore, we revealed the antimicrobial activity in HIL extract was closely associated with the induction of AMP gene expression by *L. casei* infection [25]. Therefore, this investigation suggests that although lauric acid also plays an important role in enhancing antimicrobial activity in HIL, the activity is primarily determined by the level of AMPs expression. Moreover, the present results imply that HIL extract’s antimicrobial activities can be effectively induced from HIL fed bacteria-containing feed if the lipids in HIL are fully eliminated.

We showed antimicrobial activity of the extract of HIL infected by *L. casei* using a fine needle against pathogens [21,25]. In this investigation, we found defatted HIL extract prepared from crude-oil-extracted crushed HIL powder produced from HIL fed organic waste containing F-EM also effectively suppressed pathogen growth with lower antimicrobial activity than *Lactobacillus* species (Figure 6A). In previous research, we developed a crude-oil-extraction system for HIL, and the result suggested that crude-oil-extracted crushed HIL powder by expeller extractor should be used as animal feed without further processing [33]. Here, we successfully extracted an antimicrobial substance from expeller-extractor-defatted, crushed HIL powder produced from HIL fed F-EM-containing organic wastes. Furthermore, the extracts did not affect normal animal cell survival (Figure 6B). Consequently, the results demonstrate that HIL fed F-EM-containing organic wastes should be used as sources of biodiesel, and the crude-oil-extracted crushed HIL powder could be applied as a natural antibiotic, preservative, and animal-health-friendly feed.

The industrial use of HIL as an eco-friendly fertilizer, animal feed, and protein source has been suggested by many investigators [5,34,35,36]. HIL can also be used in organic waste recycling and treatment because HIL and its adult form can effectively bioconvert organic wastes. Moreover, the importance of HIL as a feedstock for biodiesel was demonstrated by several investigations [28,37,38,39]. Several investigations showed that HIL is a living factory that can produce natural, antimicrobial substances [25,30,40,41,42]. Here, we also showed the antimicrobial activity of defatted HIL extract prepared from crude-oil-extracted crushed HIL powder. The results suggest that crude-oil-extracted crushed HIL powder is also an economical feedstock for producing natural antibiotics. The price of 1 kg crude-oil-extracted crushed HIL powder is approximately 3.1 USD in Korea. It requires 6.7 kg of crude-oil-extracted crushed HIL powder (approximately 20.7 USD) to produce 1 kg of defatted HIL extract. In contrast, melittin (purity: >97%) is sold for 1350 USD per mg by Sigma Aldrich. As shown in Figure 3, the antimicrobial activity of 5000 µg of defatted, crushed HIL powder extract is equivalent to 19.7 µg of melittin against *E. coli* and 21.3 µg of melittin against *S. aureus*. This means that defatted, crushed HIL powder extracts equivalent to the antimicrobial activity of melittin can be produced at approximately 25-fold-lower production fees.

## 5. Conclusions

We found that defatted HIL extract has antimicrobial activities against pathogens and AMP-resistant bacteria with lower toxicity to probiotics and animal cells. The present investigation demonstrates that crude-oil-extracted crushed HIL powder can be used as an eco-friendly feedstock for feed and food additives having antimicrobial activity. Taken together, the present study supports the usefulness of HIL as an eco-friendly feedstock in the biodiesel, agricultural, food, and feed industries.

## Figures and Tables

**Figure 1 animals-12-00680-f001:**
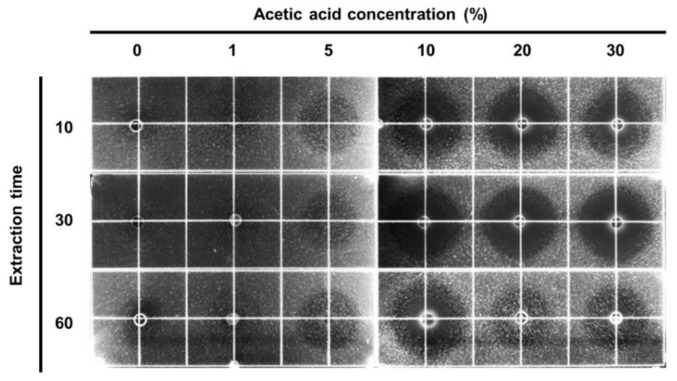
Determination of acetic acid concentration and incubation time to prepare defatted HIL extracts from crude-oil-extracted crushed HIL powder. HIL were grown for 11 days and then crude-oil-extracted HIL were prepared through the extraction of crude oil from HIL by an expeller extractor. Each batch of crude-oil-extracted, crushed HIL powder was incubated in various concentrations of acetic acid solution (0–30%) for 10, 30, or 60 min to isolate the extract. Each extract was dissolved in 0.05% acetic acid solution, and 10 µL of the extract was loaded into the wells in the underlay gel and incubated for 24 h to determine antimicrobial activity using RDA.

**Figure 2 animals-12-00680-f002:**
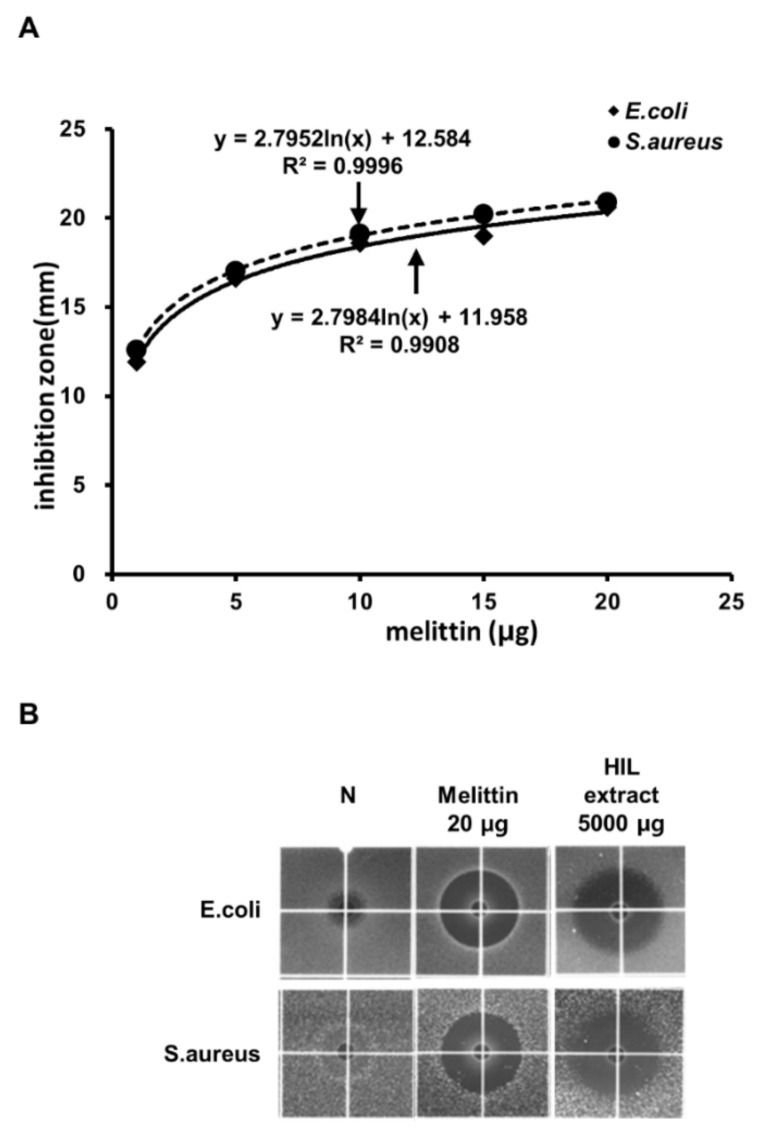
Determination of the amount of defatted HIL extract that corresponds to the antimicrobial activity of melittin. (**A**) Standard curve of the inhibitory zone with various amounts of melittin against *E. coli* and *S. aureus*. The inhibitory zone was measured as the diameter of the clear circular zone. Standard curve was presented by natural logarithmic trendline calculated by Excel in Microsoft Office 16. (**B**) Comparison of the antimicrobial activities of melittin and defatted HIL extract. Each sample was dissolved in 0.05% acetic acid solution, and 10 µL of the extract was loaded into the wells in the underlay gel and incubated for 24 h to determine antimicrobial activity using RDA. N: 0.05% acetic acid alone.

**Figure 3 animals-12-00680-f003:**
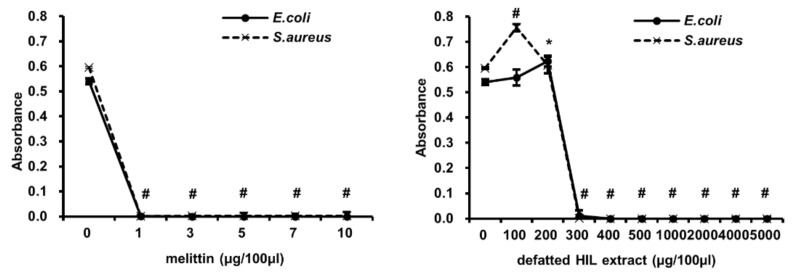
Determination of MIC of melittin and defatted HIL extract against *E. coli* and *S. aureus*. Briefly, 1 × 10^6^ cfu/mL of 90-µL *E. coli* or *S. aureus* suspension was treated with various concentrations of 10-µL melittin or defatted HIL extract dissolved in 0.05% acetic acid solution. Then, bacteria were grown for 24 h at 37 °C. The experiments were independently performed in triplicate. A concentration of 0 µg/100 µL of melittin and defatted HIL extract indicated 0.05% acetic acid alone. The statistical analysis was performed by Student *t*-test. The significant differences were determined by comparing the absorbance of each experimental group with the control (0 µg/100 µL). Each value represents the means ± standard deviations of absorbance. * and ^#^ indicate *p* < 0.05 and *p* < 0.0001, respectively.

**Figure 4 animals-12-00680-f004:**
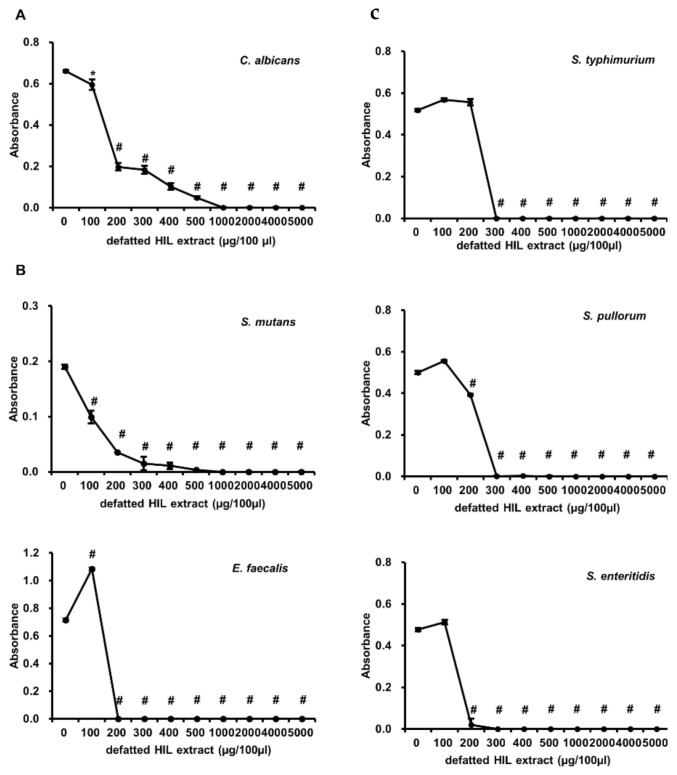
Antimicrobial activity of defatted HIL extracts against *C. albicans* (**A**), *S. mutans*, *E. faecalis*, (**B**) and *Salmonella* species (**C**). Briefly, 1 × 10^6^ cfu/mL of 90-µL *S. typhimurium*, *S. pullorum*, and *S. enteritidis*, suspensions were treated with various concentrations of 10-µL melittin or defatted HIL extract dissolved in 0.05% acetic acid solution. Then, bacteria were grown for 24 h at 37 °C. The experiments were independently performed in triplicate. A concentration of 0 µg/100 µL of melittin and defatted HIL extract indicates 0.05% acetic acid alone. The statistical analysis was performed by Student *t*-test. The significant differences were determined by comparing the absorbance of each experimental group with the control (0 µg/100 µL). Each value represents the means ± standard deviations of absorbance. * and ^#^ indicate *p* < 0.05 and *p* < 0.0001, respectively.

**Figure 5 animals-12-00680-f005:**
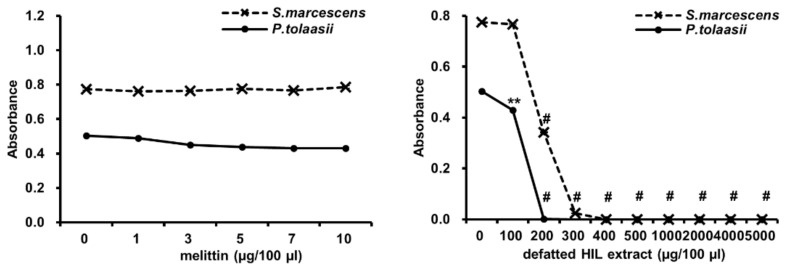
Antimicrobial activity of melittin and defatted HIL extracts against AMP-resistant *S. marcescens* and *P. tolaasii*. Briefly, 1 × 10^6^ cfu/mL of 90-µL *S. marcescens* or *P. tolaasii* suspensions were treated with various concentrations of 10-µL melittin or defatted HIL extract dissolved in 0.05% acetic acid solution. Then, bacteria were grown for 24 h at 37 °C. The experiments were independently performed in triplicate. A concentration of 0 µg/100 µL of melittin and defatted HIL extract indicates 0.05% acetic acid alone. The statistical analysis was performed by Student *t*-test. The significant differences were determined by comparing the absorbance of each experimental group with the control (0 µg/100 µL). Each value represents the means ± standard deviations of absorbance. ** and ^#^ indicate *p* < 0.05 and *p* < 0.0001, respectively.

**Figure 6 animals-12-00680-f006:**
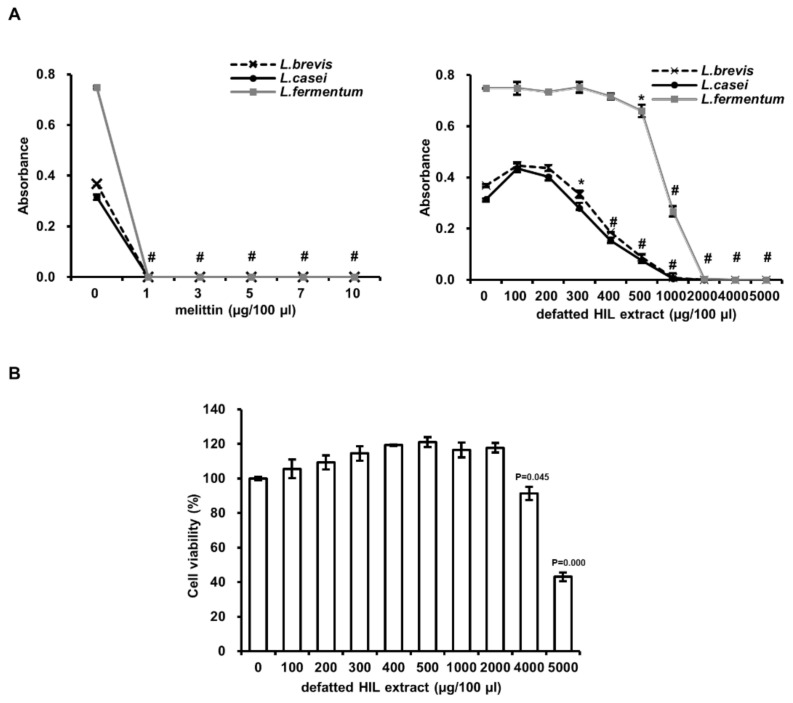
Antimicrobial activity of melittin and defatted HIL extracts against *Lactobacillus* species and cytotoxicity of defatted HIL extracts to Caco-2 human intestinal cells. (**A**) Determination of MIC of melittin and defatted HIL extract against three kinds of Lactobacillus species. Briefly, 1 × 10^6^ cfu/mL of 90-µL *E. coli* or *S. aureus* mixture was treated with various concentrations of 10-µL melittin or defatted HIL extract dissolved in 0.05% acetic acid solution. Then, bacteria were grown for 24 h at 37 °C. The experiments were independently performed in triplicate. A concentration of 0 µg/100 µL of melittin and defatted HIL extract indicates 0.05% acetic acid alone. The statistical analysis was performed by Student *t*-test. The significant differences were determined by comparing the absorbance of each experimental group with the control (0 µg/100 µL). Each value represents the means ± standard deviations of absorbance. * and ^#^ indicate *p* < 0.05 and *p* < 0.0001, respectively. (**B**) Analysis of the cytotoxic activity of defatted HIL extract. A total of 1 × 10^4^ cells/well were treated with various concentrations of defatted HIL extract dissolved in 0.05% acetic acid solution after 24 h cell attachment and then further cultured in 5% CO_2_ atmosphere for 24 h at 37 °C. The experiments were independently performed in triplicate. A concentration of 0 µg/100 µL of defatted HIL extract indicates 0.05% acetic acid alone. Bar values represent the means ± standard deviations of relative % cell viability compared to that of control (0 µg/100 µL). The significant differences were determined by comparing the % cell viability of each experimental group with that of control (0 µg/100 µL of defatted HIL extract).

## Data Availability

The data presented in this investigation are available upon request to the corresponding author.

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
