# Peer review of "Evaluation of Antimicrobial Activity in the Extract of Defatted Hermetia illucens Fed Organic Waste Feed Containing Fermented Effective Microorganisms"

_animals, 2022, doi:10.3390/ani12060680_

Round 1

Reviewer 1 Report

Review the ICZN code; genus and species should ALWAYS be written in italics, including references. Author, year (Order, Family) should be mentioned at least once in the text

Some acronyms are not comprehensive for non specialised readers. I suggest to add an appendix explaining the meaning of all the acronyms used in the text, or give the term the first time you use the acronym.

for other details see comments in the text

Author Response

Thanks for your suggestions and indications. 

We corrected manuscript based on your comments.

Reviewer 2 Report

Dear Authors

Please carefully correct the revised manuscript following the suggestion in the attached file.

Best regards,

Author Response

Thanks for your suggestions and indications. We replied to your comments as below

Title correlated and represented the whole work in this study.

Simple summary –

- HIL was represented in first time in simple summary therefore the full word with abbreviation should be represented.

: We corrected.

- Line 15: “Crushed” should be the lower case “crushed” and please correct to entitle manuscript.

: We corrected

M&M

- This study involved on several pathogenic bacteria therefore the Permission from Institutional Biosafety Committee (IBC) must be presented.

: Thanks for your comment. We performed the experiments in Biosafety Level-2 laboratory approved by Korea Disease Control and Prevention Agency. We mentioned in section 2.1.

Line 104-105 - Please provide the details on feed of HIL: Percentage of each ingredients and also chemical composition (Moisture, Crude protein, Crude fat, Crude fiber, Ash). - The diet was mixed together or separate, please explain.

: We used the mixture of dried food waste, crude chicken manure, and waste cooking oil containing fermented effective microbe solution as a feed. Therefore, we did not analyze the chemical composition. However, we mentioned the component of wastes mixture in section 2.2.

Line 105 - Can you explain in the details on “F-EM”?.

: We explain in section 2.2 in detail.

Line 106 - How you kill the HIL or just put the live HIL in microwave? Please explain more clearly.

: We describe more clearly in section 2.2.

Line 108 - In the results, you study both on the difference on acetic acid concentration and extraction time but you represented only difference on acetic acid concentration in M&M. Therefore, please clarify on the M&M.

: Thanks for your comment. We mentioned the difference of time.

Line 111 - I suggested to change for “Hold in 100C” to “Boiled”.

: We changed to “boiled”

Line 116 - How you stored the defatted HIL extract before the experiments?

: We mentioned the storage condition in section 2.3.

Line 119 - E. coli and S. aureus could be Italic.

: We corrected to Italic.

Line 128 - “venom” was duplicate, please correct.

: Thanks for your comment. We miswrote the word. We did not use purified venom as positive control. We erased the word.

Line 150 - As you used One-way ANOVA, please presentation the method to evaluate normal distribution and homogeneity of variance.

- Moreover, please identify which is fixed factor (independent variable) and dependent variable which was analyzed by One-way ANOVA because I did not find where you used One-way ANOVA.

: We mentioned details in section 2.7 and Figure captions.

 - In Figure 2A, you performed the graph and also the equation. Please explain the details that how you performed?

: We explain the details in Figure caption.

Results

- In the result could not contain reference for discussion. Because the result section could contain only the results from this study without any discussion. Please correct this entitle the result section.

Figure 1 - The unit of extraction time could be presented.

Line 155 - “Crud…” could be “Crude” please check.

 Figure 2 - Please explain what is “N”.

: We mentioned what is “N”.

Figure 6 - Please explain “Why cell viability is higher than 100%?”.

: Thanks for your comments. We don’t know why the cell viability is higher than 100%. However, it means that the extract triggers animal cell growth. Therefore, we can postulate that the extract may contain nutrients to enhance cell growth because the nutrients can be extracted from crude-oil extracted HIL crush powder.

– Figure 6B, Which statistical was performed. If it is One-way ANOVA please used the superscript to represent the statistical significance instead * or #.

: We changed the figure.

- Line 267: CO2 could be change to CO2

: We changed “CaCo-2” to “Caco-2”

Discussion

- Please do not replicate the result again in the discussion.

: Thanks for your comment. We removed replicated result in the discussion section.

Line 282 - L. casei could be italic.

: We corrected

Conclusion

- Conclusion could be revised which should contain the major result in this study with further suggestion based on the study results

: We revised conclusion.

Reviewer 3 Report

Although many experiments are performed in this paper, I have concerns about “No antimicrobial activity was observed in the extract prepared from autoclaved water-treated crushed HIL powder”. So if the antimicrobial activity is linked to the acetic acid the entire work loses the whole meaning, unless the authors succeed in improving the paper. Indeed, in the experiments performed in paragraph “3.2. Quantification of antimicrobial substance in defatted HIL extract”, “3.3. Effect of defatted HIL extract on pathogen bacterial survival”, 3.4. Effect of DFW/CM/WCO/F-EM on HIL growth” and “3.5. Effect of defatted HIL extract on Lactobacillus speies and normal animal cell survival” must be repeated analyzing the effect of acetic acid alone and comparing it to the effect of the H. illucens extracts.

Please pay attention to the italic name of bacterial species, sometimes names are not in italic.

SIMPLE SUMMARY

Line 15 Write “Production”

ABSTRACT

Line 20 write “illucens”

Line 23, 33 write “containing”

KEYWORDS

Please delete “Antimicrobial activity” and “Organic waste” that are already in the title and “Biodiesel” as it is not the main topic.

INTRODUCTION

Line 45 write “illucens”

Line 50 write “has increased”

Line 58-60: “Many researchers have investigated to search natural antibiotic substances and suggested that insects are potent natural antibiotic substances because insects produce antimicrobial peptides (AMPs) via activation of the innate immune system to keep themselves from pathogen invasion [12,13]” Please, add more references (for example doi.org/10.1007/s00018-021-03784-z; doi.org/10.3389/fcimb.2021.668632)

Line 67 As a result, the innate immune response in HIL should be activated and can produce AMPs (for example doi:10.1038/s41598-020-74017-9, https://doi.org/10.3390/cimb44010001)

Line 68, 292 write “In previous works” or “previously”

Line 77 write “have been studied as”

Line 77: “HIL also have studied as a source of biodiesel because it contains high content lipids.” Please, add a specific reference for Hermetia illucens lipids (for example https://doi.org/10.3390/su131810198)

Line 82-84: Antimicrobial properties of lauric acid and its derivative, monolaurin, are very weak. The antimicrobial activity is affected by lipids and fatty acids but not only by lauric acid. I suggest to modify the phrase and add a more specific reference to explain this (for example  https://doi.org/10.3390/insects13010041)

MATERIAL AND METHOD

Line 111 write “held”

Line 133 please rephrase “rpm shaking by shaking incubator”

RESULTS

Please substitute “crud” with “crude”

Line 157-162 in these lines no results of this study are reported, please move these sentences in the discussion section.

Line 168 substitute “during in process” with “during the process”

Line 204 “3.3. Effect of defatted HIL extract on pathogen bacterial survival” why did the authors don’t also test the bee venom?

Line 210 write “Enteroccocus”

Line 228 “3.4. Effect of DFW/CM/WCO/F-EM on HIL growth” what do these acronyms stand for? It’s very difficult to understand and follow the thesis of the authors in the discussion section.

Line 230 “an overcome from the cytotoxicity of AMPs” delete

Line 245 write “species”

Line 246 write “pathogens”

Line 250 write “species were killed”

DISCUSSION

Line 292-295: “In previous, we developed crude oil extraction system from HIL and the results suggested that crude oil-extracted crushed HIL powder by expeller extractor should be used as fertilizer and animal feed without further process [26].” There are many studies about the excellent characteristics of protein or whole powder by HIL. According to you, why should it be used as fertilizer? It’s a Strange application.

Line “Consequently, the results demonstrate that HIL fed F-EM-contained organic wastes should be used as sources of biodiesel and the frass could be applied as a natural antibiotic, preservative and animal health-friendly feeds.” Why frass “could be applied as a natural antibiotic, preservative and animal health-friendly feeds”?

CONCLUSION

Please delete “Consequently”

FIGURE

In figure 2a the effect of bee venom should be included.

In figure 3,4,5,6a the statistical analysis should be included.

FIGURE CAPTIONS

Figure 4: write “typhimurium”

Figure 6 write “Bar values represent the means ± standard deviation of relative % cell viability compared”.

Author Response

Thanks for your suggestions and indications. We replied to your comments as below.

Although many experiments are performed in this paper, I have concerns about “No antimicrobial activity was observed in the extract prepared from autoclaved water-treated crushed HIL powder”. So if the antimicrobial activity is linked to the acetic acid the entire work loses the whole meaning, unless the authors succeed in improving the paper. Indeed, in the experiments performed in paragraph “3.2. Quantification of antimicrobial substance in defatted HIL extract”, “3.3. Effect of defatted HIL extract on pathogen bacterial survival”, 3.4. Effect of DFW/CM/WCO/F-EM on HIL growth” and “3.5. Effect of defatted HIL extract on Lactobacillus species and normal animal cell survival” must be repeated analyzing the effect of acetic acid alone and comparing it to the effect of the H. illucens extracts.

: Thank you for your comments. We prepared HIL extract using 0-30% acetic acid. However, to detect the activity, we removed acetic acid completely using vacuum dryer from the extracts and all extracts were resolved by aqueous 0.05% acetic acid solution. The acetic acid concentration in Figure 1 represents the concentration used for extraction process, not the concentration of the solution to resolve. Therefore, we did not confirm the effect of aqueous 0.05% acetic acid solution in further experiments because all extracts were resolved by aqueous 0.05% acetic acid solution. Furthermore, Figure 1 shows that 0.05% acetic acid has no antimicrobial activity.

We already mentioned in the caption of Figure 1.

Please pay attention to the italic name of bacterial species, sometimes names are not in italic.

SIMPLE SUMMARY

Line 15 Write “Production”

: We corrected to “Production”

ABSTRACT

Line 20 write “illucens” : We corrected to” illucens”

Line 23, 33 write “containing”: We corrected “contained” to “containing”

KEYWORDS

Please delete “Antimicrobial activity” and “Organic waste” that are already in the title and “Biodiesel” as it is not the main topic.

: We corrected keywords

INTRODUCTION

Line 45 write “illucens”: We corrected to “illucens”

Line 50 write “has increased”: We corrected to “has increased”

Line 58-60: “Many researchers have investigated to search natural antibiotic substances and suggested that insects are potent natural antibiotic substances because insects produce antimicrobial peptides (AMPs) via activation of the innate immune system to keep themselves from pathogen invasion [12,13]” Please, add more references (for example doi.org/10.1007/s00018-021-03784-z; doi.org/10.3389/fcimb.2021.668632)

: We inserted more references

Line 67 As a result, the innate immune response in HIL should be activated and can produce AMPs (for example doi:10.1038/s41598-020-74017-9, https://doi.org/10.3390/cimb44010001)

: We inserted more references

Line 68, 292 write “In previous works” or “previously”

: We corrected to “In previous works”

Line 77 write “have been studied as”

: We corrected to “have been studied as”

Line 77: “HIL also have studied as a source of biodiesel because it contains high content lipids.” Please, add a specific reference for Hermetia illucens lipids (for example https://doi.org/10.3390/su131810198)

: We inserted more reference

Line 82-84: Antimicrobial properties of lauric acid and its derivative, monolaurin, are very weak. The antimicrobial activity is affected by lipids and fatty acids but not only by lauric acid. I suggest to modify the phrase and add a more specific reference to explain this (for example  https://doi.org/10.3390/insects13010041)

: We modified the phrase and inserted specific reference.

MATERIAL AND METHOD

Line 111 write “held”

: We corrected to “held”

Line 133 please rephrase “rpm shaking by shaking incubator”

: We removed “by shaking incubator”.

RESULTS

Please substitute “crud” with “crude”

Line 157-162 in these lines no results of this study are reported, please move these sentences in the discussion section.

: Thanks for you comments. We removed the sentences from “result section”. But we did not move the sentences because similar sentences were mentioned in “discussion section“.

Line 168 substitute “during in process” with “during the process”

Line 204 “3.3. Effect of defatted HIL extract on pathogen bacterial survival” why did the authors don’t also test the bee venom?

: Thanks for your comment. We used melittin as positive control. Melittin is a major component of bee venom and microbial activity of bee venom is predominantly determined by melittin. Therefore, we didn’t assess antimicrobial activity of bee venom.

We removed miswritten “purified been venom” in material and method and results sections.

Line 210 write “Enteroccocus”

: We corrected to “Enteroccocus”

Line 228 “3.4. Effect of DFW/CM/WCO/F-EM on HIL growth” what do these acronyms stand for? It’s very difficult to understand and follow the thesis of the authors in the discussion section.

: Thanks for your comment. The title was mis-written. We corrected the title to “Effect of defatted HIL extract on AMP-resistant bacterial survival”

Line 230 “an overcome from the cytotoxicity of AMPs” delete

: Thanks for your comment. It means AMP-resistant bacteria. Therefore, to describe the purpose of this experiment, the sentence must be mentioned.

Line 245 write “species”

: We corrected to “species”

Line 246 write “pathogens”

: We corrected to “species”

Line 250 write “species were killed”

: We corrected to “species were killed”

DISCUSSION

Line 292-295: “In previous, we developed crude oil extraction system from HIL and the results suggested that crude oil-extracted crushed HIL powder by expeller extractor should be used as fertilizer and animal feed without further process [26].” There are many studies about the excellent characteristics of protein or whole powder by HIL. According to you, why should it be used as fertilizer? It’s a Strange application.

: Thanks for your comment. We agree your suggestion. We removed “fertilizer”.

Line “Consequently, the results demonstrate that HIL fed F-EM-contained organic wastes should be used as sources of biodiesel and the frass could be applied as a natural antibiotic, preservative and animal health-friendly feeds.” Why frass “could be applied as a natural antibiotic, preservative and animal health-friendly feeds”?

: Thanks for your comment. We missed to correct the word. So, we corrected “frass” to “and the crude oil-extracted crushed HIL powder”.

CONCLUSION

Please delete “Consequently”

: We erased the word.

FIGURE

In figure 2a the effect of bee venom should be included.

: We mis-inserted vee venom. We used only melittin as a positive control because melittin is a major component in purified bee venom.

In figure 3,4,5,6a the statistical analysis should be included.

: We showed the significance differences in each figure.

FIGURE CAPTIONS

Figure 4: write “typhimurium”

: We corrected

Figure 6 write “Bar values represent the means ± standard deviation of relative % cell viability compared”.

: We corrected.

Reviewer 4 Report

The manuscript submitted for publication to Animals is really interesting. Hermetia illusens is an insect larva with great potential. Cytotoxicity assays, necessary in this type of assay, are appropriate.

Small objections to remedy:

- M&M's. Section 2.1. It would be interesting to separate pathogenic bacteria from those that are not (prebiotics), as well as from the yeast C. albicans

- Section 2.4 and following. Italicize the names of microorganisms.

- Would it be possible to reduce panels in figure 4? Grouping according to what was said for section 2.1

Therefore, the results demonstrate that crude-oil Crushed HIL powder prepared from HIL fed organic waste contained fermented effective microorganisms for biodiesel production should be used as the feedstock of synthetic preservative-free livestock feed and food additives. Taken together, the present study supports the usefulness of HIL as eco-friendly feedstock in biodiesel, agricultural, food, and feed industries.

I recommend to publish this manuscript after minor amends.

Author Response

Thanks for your suggestions and indications. We replied to your comments as below.

The manuscript submitted for publication to Animals is really interesting. Hermetia illusens is an insect larva with great potential. Cytotoxicity assays, necessary in this type of assay, are appropriate.

Small objections to remedy:

- M&M's. Section 2.1. It would be interesting to separate pathogenic bacteria from those that are not (prebiotics), as well as from the yeast C. albicans

: Thanks for your comment. We corrected.

- Section 2.4 and following. Italicize the names of microorganisms.

: We corrected.

- Would it be possible to reduce panels in figure 4? Grouping according to what was said for section 2.1

: Thanks for your comment. We re-edited figure 4 and reduced the panels.

Round 2

Reviewer 2 Report

Dear Authors

The correction was performed according to the previous revision. Therefore, I did not have any suggestion or correction.

Best regards

Reviewer

Author Response

Thanks for your comments.

We corrected miswritten spells.

Reviewer 3 Report

MATERIAL AND METHODS

-Did the authors quantitatively analyze the substance that should be totally fat-free? If these analyses have not been done, it would be advisable to check the % of lipids in the defatted extract. If it is totally absent, the work can be considered acceptable otherwise it should be redone.

-Why the authors dissolved the extracts specifically in acetic acid.

-Once the authors tested the acid acetic concentrations up to 30%, what is the real meaning of writing “the highest antimicrobial activity was shown when crushed HIL powder was treated with 20% acetic acid solution at 100℃ for 30 min” as probably the activity is linked to acetic acid?

As the antimicrobial activity could be related to acetic acid, it’s better to have an image (Figure 2) in which there is only acetic acid without the Hermetia illucens extract, to compare the antimicrobial activity of extract+acetic acid with acetic acid alone. Once the authors have added the acetic acid alone, the choice to use 0.05% of acetic acid could be explained.

- Moreover, as I have already suggested, each experiment performed in paragraph “3.2. Quantification of antimicrobial substance in defatted HIL extract”, “3.3. Effect of defatted HIL extract on pathogenic bacterial survival”, 3.4. Effect of defatted HIL extract on AMP-resistant bacterial survival” and “3.5. Effect of defatted HIL extract on Lactobacillus species and normal animal cell survival” must be repeated analyzing the effect of acetic acid alone and comparing it to the effect of the H. illucens extracts.

DISCUSSION and CONCLUSION

“Here, we showed the highest antimicrobial activity was observed when crude-oil extracted crushed HIL powder was extracted by diluted aqueous acetic acid solution”

“We found that the extract of crude oil-extracted crushed HIL powder has antimicrobial activities against pathogens and AMP-resistanct bacteria with the lower toxicity to 336 probiotics and animal cells”.

These statements are not in disagreement with what has been said so far, in which “defatted Hermetia illucens extract” has been mentioned.

Author Response

Thanks for your suggestions and indications. We replied to your comments as below.

-Did the authors quantitatively analyze the substance that should be totally fat-free? If these analyses have not been done, it would be advisable to check the % of lipids in the defatted extract. If it is totally absent, the work can be considered acceptable otherwise it should be redone.

: Thanks for your comment. The lipid content in HIL and crushed HIL powder were 86.66±1.54% and 7.17±0.73 %, respectively. We showed the lipid contents as a supplementary information.

-Why the authors dissolved the extracts specifically in acetic acid.

: We extracted 20% acetic acid solution. Then, removed the solution by using vacuum drier. Dried extract was dissolved in 0.05% acetic acid because the solubility of the extract was higher in 0.05% acetic solution than in distilled water. However, 0.05% acetic acid solution didn’t affect to antimicrobial activity of the extract.

-Once the authors tested the acid acetic concentrations up to 30%, what is the real meaning of writing “the highest antimicrobial activity was shown when crushed HIL powder was treated with 20% acetic acid solution at 100℃ for 30 min” as probably the activity is linked to acetic acid?

: The activity is not linked to acetic acid. During the extraction, we heated crushed HIL powder and treated acetic acid. Moreover, the activity of AMP in the extract should be affected by heating time and acetic acid solution. Therefore, we performed the experiment to optimize the extraction condition.

As the antimicrobial activity could be related to acetic acid, it’s better to have an image (Figure 2) in which there is only acetic acid without the Hermetia illucens extract, to compare the antimicrobial activity of extract+acetic acid with acetic acid alone. Once the authors have added the acetic acid alone, the choice to use 0.05% of acetic acid could be explained.

: Thanks for your comment. All samples are dissolved by 0.05% acetic acid solution and the acetic acid solution was loaded onto the gel as a control, mentioned as N. We mentioned “N” as “not treated”. We corrected “not treated” to “0.05% acetic acid alone” to clearly describe that 0.05% acetic acid was used in all experiment.

- Moreover, as I have already suggested, each experiment performed in paragraph “3.2. Quantification of antimicrobial substance in defatted HIL extract”, “3.3. Effect of defatted HIL extract on pathogenic bacterial survival”, 3.4. Effect of defatted HIL extract on AMP-resistant bacterial survival” and “3.5. Effect of defatted HIL extract on Lactobacillus species and normal animal cell survival” must be repeated analyzing the effect of acetic acid alone and comparing it to the effect of the H. illucens extracts.

: Thanks for your comment. All 0 µg/ml of HIL extract samples are used as control and prepared with 0.05% acetic acid solution alone. We mentioned about that in Figure captions.

DISCUSSION and CONCLUSION

“Here, we showed the highest antimicrobial activity was observed when crude-oil extracted crushed HIL powder was extracted by diluted aqueous acetic acid solution”

“We found that the extract of crude oil-extracted crushed HIL powder has antimicrobial activities against pathogens and AMP-resistanct bacteria with the lower toxicity to 336 probiotics and animal cells”.

These statements are not in disagreement with what has been said so far, in which “defatted Hermetia illucens extract” has been mentioned.

: Thanks for your comment. We corrected the sentences.